# Cognitive function in non-hospitalized patients 8–13 months after acute COVID-19 infection: A cohort study in Norway

Knut Stavem[1,2,3]*, Gunnar Einvik[1,3], Birgitte Tholin[3,4], Waleed Ghanima[3,4], Erik Hessen[5,6], Christofer Lundqvist[3,5]

**1** Department of Pulmonary Medicine, Division of Medicine, Akershus University Hospital, Nordbyhagen, Norway, **2** Health Services Research Unit, Akershus University Hospital, Nordbyhagen, Norway, **3** Institute of Clinical Medicine, University of Oslo, Oslo, Norway, **4** Clinic of Internal Medicine, Østfold Hospital, Grålum, Norway, **5** Department of Neurology, Division of Medicine, Akershus University Hospital, Nordbyhagen, Norway, **6** Department of Psychology, University of Oslo, Oslo, Norway

* knut.stavem@medisin.uio.no

**Data Availability Statement:** The ethical approval granted by the Regional Committees for Medical and Health Research Ethics in Norway does not allow public sharing of the data. A data set can be

## Abstract

Studies have reported reduced cognitive function following COVID-19 illness, mostly from hospital settings with short follow-up times. This study recruited non-hospitalized COVID-19 patients from a general population to study prevalence of late cognitive impairment and associations with initial symptoms. We invited patients with PCR-confirmed COVID-19. A postal questionnaire addressed basic demographics, initial COVID-19 symptoms and co-morbidity about 4 months after diagnosis. About 7 months later, we conducted cognitive tests using the Cambridge Neuropsychological Test Automated Battery, comprising four tests for short-term memory, attention and executive function. We present descriptive statistics using z-scores relative to UK population norms and defined impairment as z-score <-1.5. We used multivariable logistic regression with impairment as outcome. Continuous domain scores were analysed by multiple linear regression. Of the initial 458 participants; 305 were invited, and 234 (77%) completed cognitive testing. At median 11 (range 8–13) months after PCR positivity, cognitive scores for short term memory, visuospatial processing, learning and attention were lower than norms (p≤0.001). In each domain, 4–14% were cognitively impaired; 68/232 (29%) were impaired in ≥ 1 of 4 tests. There was no association between initial symptom severity and impairment. Multivariable linear regression showed association between spatial working memory and initial symptom load (6–9 symptoms vs. 0–5, coef. 4.26, 95% CI: 0.65; 7.86). No other dimension scores were associated with symptom load. At median 11 months after out-of-hospital SARS-Cov-2 infection, minor cognitive impairment was seen with little association between COVID-19 symptom severity and outcome.

made available for scientific analysis on request, provided that the respective research institution proofs handling of the data strictly in accordance with ethical regulations (written ethics protocol, full compliance with the Declaration of Helsinki). To ensure full anonymity only the main variables of the final analyses are provided. We confirm that the data file provided constitutes the minimal data set necessary to replicate the findings of our study in their entirety. Data requests can be made to corresponding author, Knut Stavem (knut. stavem@medisin.uio.no), or data custodian, Haldor Husby (haldor.husby@ahus.no). Contact information to the Regional Committee for Medical and Health Research Ethics: REC south-east-Secretariat, email: post@helseforskning.etikkom. no.

**Funding:** The authors received no specific funding for this work.

**Competing interests:** The authors have declared that no competing interests exist.

## Introduction

In recovery after acute coronavirus disease 2019 (COVID-19), patients commonly complain about disturbed concentration or memory, insomnia, or fatigue [1–3]. Patients with critical illness, such as acute respiratory distress syndrome (ARDS), similarly have problems with memory, attention, concentration and/or global loss of cognitive function, often persisting after 1 year [4]. There is fear of long-term neurocognitive deficits following COVID-19, possibly due to hypoxemia or cerebral hypoxia, metabolic dysfunctions, viral encephalitis, immune activation, or other mechanisms [5, 6].

A recent review of 12 studies with objective neuropsychological test data of patients with recent COVID-19 reported that global cognitive impairment, impairment in memory, attention and executive function, and verbal fluency were common [7]. These studies most often used global cognitive tests designed for neurodegenerative disorders, such as the Montreal Cognitive Assessments (MoCa), and there was wide variation in populations, e.g. hospitalized patients vs. non-hospitalized, severity of acute COVID-19, ventilator use, follow-up time, age groups, and assessment methods [8–10].

Many patients would be expected to improve during the first 6–12 months, and it is unknown if earlier assessments are good indicators of the long-term impact on cognitive function. There is little detailed information on cognitive function after recovery in non-hospitalized patients, who constitute the majority of those affected by COVID-19. Furthermore, it is unascertained if any persistent neurocognitive impairment is related to the severity of COVID-19, and, if so, the potential mechanism. Two recent large studies comprising hospitalized and non-hospitalized patients 2–8 months after COVID-19 reported that patients having recovered from COVID-19 performed worse on a range of comprehensive cognitive tests than population norms or control subjects, with significantly more cognitive impairment in those hospitalized [11, 12].

This study aimed to determine the prevalence of objective cognitive deficits in a geographical cohort of well-characterized non-hospitalized patients 8–13 months after COVID-19. In addition, we wanted to investigate variables associated with neurocognitive deficits with a special focus on symptoms suggestive of central nervous system (CNS) engagement during the acute phase of COVID-19.

## Methods

### Study design and sample

This study was a longitudinal study in a geographical cohort in the catchment areas of two Norwegian hospitals, Akershus University Hospital and Østfold Hospital. The hospitals have a combined catchment area of about 900,000 inhabitants from both urban and rural areas, comprising about 17% of the population of Norway. For this study, we initially invited subjects ≥18 years with a positive polymerase chain reaction (PCR) for severe acute respiratory syndrome coronavirus-2 (SARS-CoV-2) from the microbiology laboratories of the two hospitals and the largest private microbiology laboratory in the geographical area, Fürst Medical Laboratory, until June 1, 2020. We anticipated that these labs would have analyzed more than 90% of the SARS-CoV-2 tests in the hospitals' catchment areas. Patients were eligible if they were alive and had not been hospitalized for COVID-19, defined as admitted to hospital <22 days after the positive PCR test. We excluded patients for the following reasons: living outside the hospitals' catchment areas, no valid 11-digit Norwegian national identity number, or permanent address in a nursing home and dementia (Fig 1).

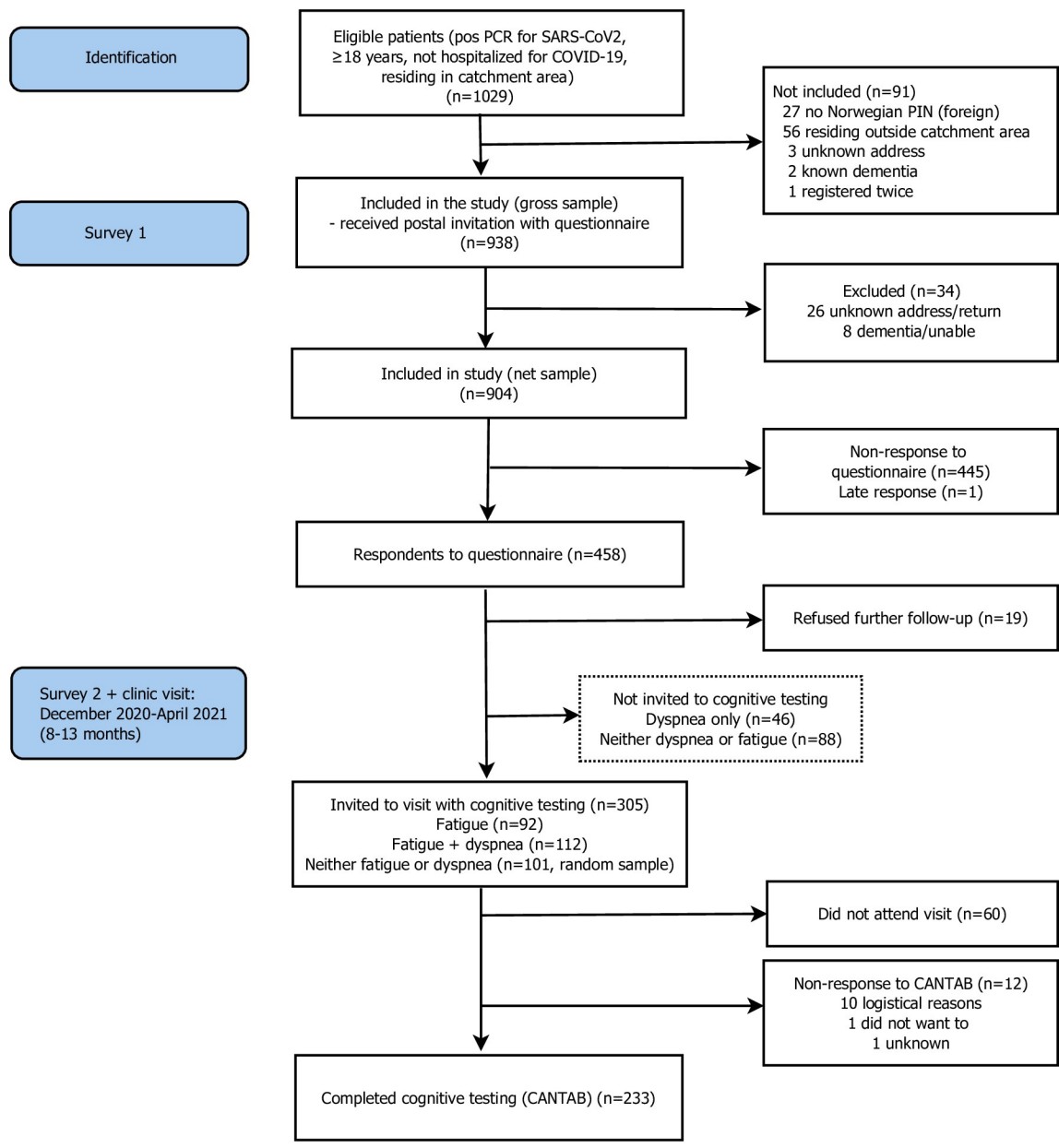

**Fig 1. Flow chart of study recruitment and attrition.**

The study participants were recruited during the first wave of COVID-19 in Norway until June 1, 2020, when a total of 8,443 laboratory-confirmed cases had been reported, or 157 cases per 100,000 inhabitants, and 1,068 patients with COVID-19 had been hospitalized [13]. Based on a limited national sample, the dominating subtype during this period was B.1, and less commonly B.1.1 and B.1.1.1 [14].

## Initial mixed mode survey

The initial survey was a postal survey at the end of June 2020 providing study information and a consent form to eligible subjects, about 1 to 4 months after contracting COVID-19.

Participants could respond to an enclosed paper questionnaire or to a similar web-based questionnaire [15]. Following exclusions, we invited 938 subjects to participate in the survey, and 458 responded after one postal reminder.

The questionnaire contained background variables, symptoms during acute COVID-19 and prevailing symptoms at the time of the survey, as well as several standard questionnaires, including the EQ-5D-5L health status questionnaire [16] and Chalder fatigue questionnaire (CFQ) [17].

### Follow-up survey and visit

Between December 2020 and April 2021 we invited respondents in the first survey, excluding those declining further participation (n = 19), to a follow-up visit including blood samples and more questionnaires. To enable differentiating the clinical examination program during the follow-up visits, we divided all participants into three symptomatic groups according to their report of dyspnea (modified Medical Research Council score $\geq$1) (n = 46), fatigue (CFQ score $\geq$4) (n = 92), dyspnea and fatigue (n = 112) in the first survey, and a random sample of 101 participants with no dyspnea or fatigue (n = 189). The fatigue, dyspnea and fatigue, and the no dyspnea or fatigue groups were invited to cognitive testing (n = 305) (Fig 1).

### Variables from the initial survey

From the initial survey, we used information on demographics, education, comorbidity, smoking status, height, weight, initial COVID-19 symptoms, and date of first symptom.

*Comorbidity* was recorded using a checklist of 21 diseases and conditions, 18 of which constituted a self-report version of the Charlson comorbidity index [18] and some additional items that we assumed may be related to COVID-19. We summed up the number of comorbidities and categorized this comorbidity index as 0, 1, 2, $\geq$3 comorbidities.

*Symptoms* during the acute phase of COVID-19 and at the time of survey were assessed using a checklist of 23 self-reported symptoms [15]. The participants could check for having the symptom during COVID-19 (yes/now) and if they still had the symptom (yes). To denote the severity of the acute COVID-19 infection in the current analysis, we used the number of retrospectively reported symptoms during the acute disease, categorized into tertiles (0–5, 6–9, 10–23) [15].

*Symptoms of anxiety/depression* were extracted from scores on the EQ-5D-5L questionnaire [16] at 3 months and dichotomized as 0 (1 none/2 slight) or 1 (3 moderate/4 severe). None of the respondents had used the response option 5 (extreme).

### Assessment of cognitive function

**Selection of tests.**   We searched for brief test batteries for assessing cognitive function because of perceived time constraint during the visits and fatigue among the participants. They should have available validated norms and be possible to be administered by paraprofessionals or technically trained staff.

For this study, we focused on selected cognitive domains: short-term memory, attention and executive function. We chose a battery of tasks from the Cambridge Neuropsychological Test Automated Battery (CANTAB, Cambridge Cognition Ltd, Cambridge, UK) [19], using an iPad (Apple Inc., Cupertino, CA, United States). The CANTAB is widely used, offers many neurocognitive tasks, and has been validated in a variety of neurological and psychiatric conditions, including in Norway [20, 21]. We selected one warm-up task, a motor screening test (MOT), and four tests: (1) Delayed matching to sample (DMS), testing short term memory, visuospatial processing, learning and attention; (2) One-touch Stockings of Cambridge (OTS),

**Table 1. Cognitive tests and key outcomes.**

| Test | Domain | Time (min) | Key outcomes |
|---|---|---|---|
| *Delayed matching to sample (DMS)* | | | |
| The participant is shown a complex visual pattern, followed by four similar patterns, after a short delay. The participant selects the pattern that exactly matches the sample. The delay varies during the test. | Visuospatial processing, attention, and short- term memory. | 7 | DMS-PCAD: Percent correct responses on first choice in all tasks with delay |
| *Rapid visual information processing (RVP) with three targets* | | | |
| Digits from 2 to 9 are presented successively at the rate of 100 digits per minute and in a pseudorandom order. Participants respond to specified sequences of digits, with increasing difficulty. | Sustained attention. | 9 | A' (A prime): a signal detection measure of response sensitivity to the target, regardless of response tendency (expected range is 0–1); The median response latency |
| *One-touch stockings of Cambridge (OTS) standard version* | | | |
| The screen shows 2 displays, each with 3 colored balls. The participant must determine the minimum number of moves to transfer from the starting configuration to the target configuration, given specific rules/restriction on moves. | Executive function, spatial planning and working memory. | 10 | OTS-PSFC: No. of OTS problems solved on first choice. |
| *Spatial working memory (SWM) standard 2.0 extended* | | | |
| Participants click on colored boxes presented on the screen. They inspect their contents, reveal a token hidden below, and move the token to a collection area. | Working memory and strategy. Retention and manipulation of visuospatial information. | 6 | SWM between errors: No. of times the participant incorrectly revisits a box, calculated across all assessed 4, 6, and 8 token trials. |

testing executive function, which includes high level thinking and decision-making processes such as mental flexibility, planning and problem solving; (3) Rapid visual information processing (RVP), testing sustained attention; (4) Spatial working memory (SWM), testing working memory and strategy. The battery was expected to take 34 min to complete. All tests were available in Norwegian and English and had UK population norms from the vendor for subjects aged 18–85+ years, except for the MOT [22].

The tested cognitive processes do not work in isolation. Therefore, it may be beneficial to include tests which combine domains, as well as tests targeting separate domains [19]. Details of the four tests with available norm values are provided in Table 1.

**Procedure.** The patients visited the outpatient clinic in 1 of the 2 hospitals, and as part of the work-up they received instructions on how to use the tablet from a study coordinator and completed the test immediately in the clinic. As a warm-up to familiarize the participants with the tablet and the system, they first completed the MOT (2 min). They then completed the RVP with three targets (9 min), DMS (7 min), OTS (10 min), and the SWM (6 min) test [22].

## Statistical analysis

Descriptive statistics are presented using the mean (SD) or median (range), as appropriate. Non-response was assess using t-test or chi-square t-test, as appropriate. As some of the tests comprised several sub-tests that may be reported separately, we present cognitive scores for the sub-test from each of the four composite tests that was most comprehensive and had a broad and continuous distribution of scores (Fig 2). The results of the four cognitive tests are reported as z-scores, i.e., the number of SDs below or above the mean score in an age, sex and education-adjusted norm population mean. We used a z-score of <-1.5 (corresponding to the 6.68-percentile of the norm distribution) to denote an abnormal score. Because some of the tests showed non-normal distributions of scores, we used Wilcoxon signed-rank test to compare scores with population norms. As elderly patients may be more vulnerable to the effects of SARS-Cov-2 virus infections, in an additional post hoc analyses, we repeated the analysis above in a subset with participants ≥60 years of age.

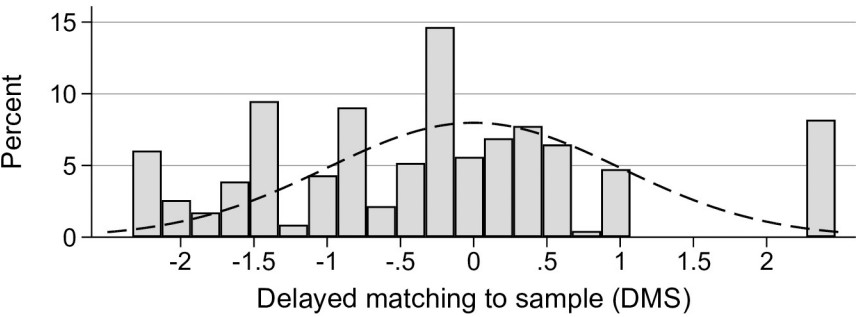

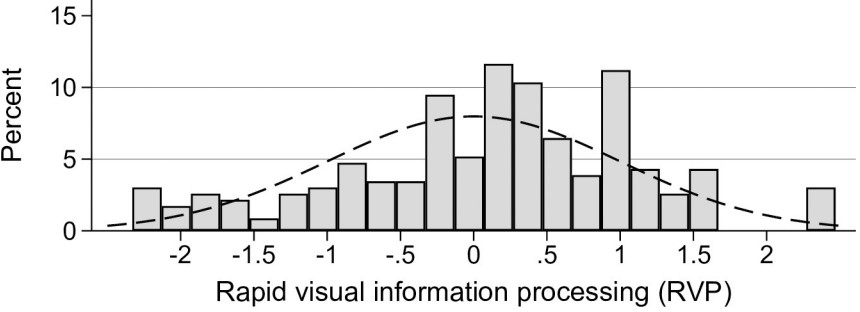

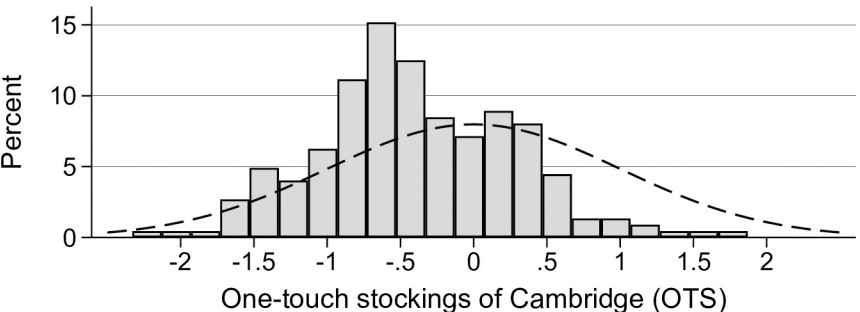

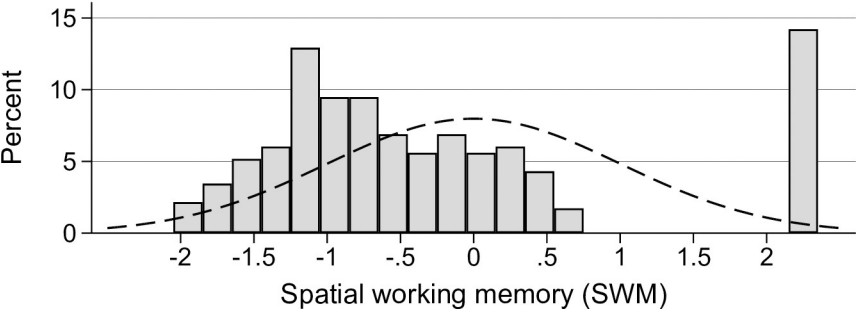

**Fig 2. Distribution of z-scores for CANTAB cognitive tests, with standard normal curve representing the norm population.**

We are not aware of a case definition for cognitive impairment after COVID-19. Previous research of critically ill patients following ICU stay used a case definition of z-score < -2.0 in 2 of 7 dimensions, or < -1.5 in 3 of 7 dimensions [23]. Because we used four tests, we determined

the proportion of patients having a z-score <-1.5 on 1 or more of 4 tests, as well as on 2 or more of 4 tests. This is similar to definitions used elsewhere [12, 24].

We assessed predictors of having a z-score <-1.5 on 1 or more of 4 tests using multivariable logistic regression analysis. Because of a limited number of events, we selected the following candidate variables prior to the analysis: age, education (four levels), Charlson comorbidity index (0, 1, 2, ≥3), number of acute COVID-19 symptoms (0–5, 6–9, 10–23), and confusion during COVID-19. We selected the latter variable because we hypothesized this to be the single symptom that was most likely to be associated with hypoxia in the acute phase.

To assess determinants of the continuous dimension scores, we analyzed the data using multiple linear regression analysis. Because of slight non-normality of the distribution of the residuals and influential outliers for several of the test, we estimated 95% CI and p-values using bootstrapping with 1000 iterations for all models. We chose candidate variables for all models based on the availability and the literature [25]: age, sex, education, born in Norway, Charlson comorbidity, headache, confusion, seizures, smell loss and aggregate number of symptoms during acute COVID-19 (0–5, 6–9, 10–23) and symptoms of anxiety/depression (yes vs. no). All variables were forced into the models without a variable selection procedure.

Because of risk of attrition affecting the outcome data, non-selection of dyspnea patients and random sampling of a subset of patients without dyspnea/fatigue, we conducted a sensitivity analysis where we weighted the dyspnea/fatigue patients with the inverse of the sampling weight, and then calculated the prevalence of cognitive z-scores <-1.5, repeated the logistic regression analysis and the multiple linear regression analyses (without bootstrapping).

We used Stata version 16.1 for statistical analysis (StataCorp., College Station, TX, United States). We chose a 5% significance level.

## Ethics

All participants gave written informed consent to participate in the study. All procedures were performed according to the Helsinki protocol and were passed by the Ethics committee of South-East Norway and by the Data inspection officer of Akershus University Hospital.

## Results

### Sample

Of 904 non-hospitalized SARS-Cov-2 positive patients invited to participate, 458 (51%) responded to the first questionnaire; 19 of these patients refused further follow-up, thus 439 were available for clinical visits. Because of stratification and selection of a random sample of those without dyspnea or fatigue after 3 months, we invited 305 of patients to the visit including cognitive testing. Of these, 245 (80%) participated, and 233 (76%) completed the CANTAB cognitive test battery (Fig 1).

The first questionnaire survey was performed a median of 4 months (118 days, range 41–193 days) after symptom onset, and cognitive tests were performed a median of 11 months (330 days, range 241–394 days) after the positive PCR. Respondents' mean age was 50.1 years (SD 14.8), and 59% were females (Table 2).

The most common initial COVID-19-associated symptoms reported were fever, headache, smell loss and dyspnea (72, 71, 68 and 65% respectively), and 16% and 6% respectively reported confusion and seizures. In total, 78% reported more than five COVID-19 symptoms during the acute phase (Table 2).

Those completing the CANTAB battery had more symptoms in the acute phase, poorer health-related quality of life (EQ-5D Index) than non-participants/non-respondents to CAN-TAB, but there was no difference in age, sex, marital status or Norwegian origin (S1 Table).

**Table 2. Descriptive statistics for subjects responding to CANTAB (N = 233).**

| | n | Freq. | Mean/Prop. | SD |
|---|---|---|---|---|
| Age, date of response | 233 | | 49.8 | 14.7 |
| Sex, males | 233 | 95 | 0.41 | |
| *Highest attained education* | 233 | | | |
| Primary school (<10 years) | | 17 | 0.07 | |
| Secondary school (10–12 years) | | 95 | 0.41 | |
| University level, < 4 years | | 60 | 0.26 | |
| University level, ≥ 4 years | | 61 | 0.26 | |
| *Marital status* | 233 | | | |
| Single | | 33 | 0.14 | |
| Married | | 109 | 0.47 | |
| Cohabiting | | 59 | 0.25 | |
| Divorced/separated | | 23 | 0.10 | |
| Widowed | | 9 | 0.04 | |
| *Working status* | 232 | | | |
| Full-time working | | 149 | 0.64 | |
| Reduced hours (<80% of full-time) | | 29 | 0.12 | |
| On sick-leave | | 18 | 0.08 | |
| Retired | | 27 | 0.12 | |
| Pupil/student | | 9 | 0.04 | |
| Born in Norway | 231 | 200 | 0.87 | |
| *Smoking status* | 231 | | | |
| Never smoked daily | | 146 | 0.63 | |
| Former daily smoker | | 71 | 0.31 | |
| Current daily smoker | | 13 | 0.06 | |
| *Infection place* | 233 | | | |
| Travel abroad | | 57 | 0.25 | |
| In Norway, known contact | | 103 | 0.44 | |
| In Norway, unknown contact | | 72 | 0.31 | |
| Body mass index (self-report), kg/m$^2$ | 232 | | 27.3 | 4.8 |
| *No. of comorbidites, categorized* | 233 | | | |
| 0 | | 108 | 0.46 | |
| 1 | | 76 | 0.33 | |
| 2 | | 29 | 0.12 | |
| ≥3 | | 20 | 0.09 | |
| Fever | 231 | | 0.71 | |
| Dyspnea | 229 | | 0.65 | |
| Confusion | 232 | | 0.16 | |
| Headache | 232 | | 0.71 | |
| Seizures | 230 | | 0.06 | |
| Smell loss | 230 | | 0.68 | |
| *No. of symptoms during acute COVID-19* | 233 | | | |
| 0–5 | | 52 | 0.22 | |
| 6–9 | | 78 | 0.33 | |
| 10–23 | | 103 | 0.44 | |
| *EQ-5D Anxiety/Depression after 3 months* | 232 | | | |
| 1–2 (None/slight) | | 208 | 0.90 | |
| 3–4 (Moderate/severe) | | 24 | 0.10 | |

*(Continued)*

**Table 2.** (Continued)

| | n | Freq. | Mean/Prop. | SD |
|---|---|---|---|---|
| Time from symptom start to response survey 1, days. | 229 | | 117 | 26 |
| Median (range) | 229 | | 118 | 41 to 193 |
| Time from PCR test to cognitive test, days. | 232 | | 330 | 27 |
| Median (range) | 232 | | 330 | 241 to 394 |

## Cognitive assessments

Tests of short-term memory, visuospatial processing, learning and attention (SWM, DMS and RVP) showed small reductions in cognitive scores about 11 months after SARS-CoV-2 PCR positivity compared to UK population norms (p≤0.001), after adjustment for age, sex and education. There was, in the total sample, no reduction for executive function (OTS), p = 0.167) (Table 3). The distribution of z-scores on the chosen cognitive tests are shown in Fig 2.

In the *post hoc* analysis of those ≥ 60 years, findings were similar as in the total sample, except for OTS scores where elderly individuals were more reduced and scored below norms (p = 0.030) (S2 Table).

Using a definition of impaired cognition of >1.5 SD below norms, only a small percentage of patients (4–14%) were cognitively impaired at follow-up (Table 4). Using the same 1.5 SD threshold, 29% (68/232) of participants had an impaired cognitive function in at least 1 of 4 tests and 6% (15/232) in at least 2 of 4 tests. Multivariable logistic regression using impairment of more than 1.5 SD in at least 1 of 4 dimensions and adjusting for age, education and number of comorbidities, showed no association with total COVID-19-associated symptoms nor with confusion as initial symptom.

In our adjusted, multivariable, linear regression analysis (Table 5), cognitive score on the SWM test, was the only test result significantly associated with the number of initial COVID-19 symptoms. DMS, OTS, and RVP (representing visuospatial processing, executive function and sustained attention respectively) were not associated with the total number of COVID-19 symptoms. None of the cognitive tests showed significant association with individual, CNS-related symptoms such as headache, confusion, seizures and loss of smell during COVID-19.

**Table 3. Cognitive test scores and comparison with norm population.**

| Test | Description | Score (range) | n | Raw score Mean | SD | n | Z-score* Mean | SD | P** | Z-score <-1.5 No. | % |
|---|---|---|---|---|---|---|---|---|---|---|---|
| Delayed matching to sample (DMS) | DMS Percent Correct (All trials containing a delay) | 0 to 100 (best) | 233 | 81.3 | 11.8 | 232 | -0.31 | 1.16 | <0.001 | 33 | 14.2 |
| One Touch Stockings of Cambridge (OTS), standard version | No. of OTS Problems Solved on First Choice | 0 to 15 (best) | 233 | 10.3 | 3.1 | 232 | 0.09 | 1.03 | 0.167 | 22 | 9.5 |
| Rapid visual information processing (RVP), 3 targets | RVP A' (A prime) measures a subject's sensitivity to the target sequence (string of three numbers), regardless of response tendency. | 0 to 1 (best) | 225 | 0.89 | 0.05 | 224 | -0.39 | 0.67 | <0.001 | 9 | 4.0 |
| Spatial working memory (SWM), recommended standard 2.0 extended | SWM Between Errors. No. of times the subject incorrectly revisits a box in which a token has previously been found. Across all assessed trials. | 0 to 153 (worst) | 233 | 12.9 | 9.5 | 232 | -0.27 | 1.23 | 0.001 | 22 | 9.5 |

* higher z-score is better performance

** Z-score comparison with norm population (mean = 0)

**Table 4. Determinants of cognitive impairment (z-score <-1.5) in at least 1 of 4 tests.** Logistic regression analysis (n = 232).

| Covariate | N | Odds ratio | 95% Conf. Interval | P |
|---|---|---|---|---|
| Age, per decade | 232 | 1 | (0.80 to 1.26) | 0.98 |
| Education | | | | |
| Primary school (<10 years) | 17 | 1 | | |
| Secondary school (10–12 years) | 94 | 0.74 | (0.24 to 2.29) | 0.60 |
| University, <4 years | 60 | 0.61 | (0.18 to 2.07) | 0.43 |
| University, ≥4 years | 61 | 0.78 | (0.24 to 2.52) | 0.68 |
| No. of comorbidities, categorized | | | | |
| 0* | 108 | 1 | | |
| 1 | 76 | 0.82 | (0.40 to 1.64) | 0.57 |
| 2 | 29 | 1.93 | (0.76 to 4.92) | 0.168 |
| ≥3 | 19 | 0.79 | (0.23 to 2.70) | 0.71 |
| Confusion | | | | |
| No* | 195 | 1 | | |
| Yes | 37 | 1.22 | (0.53 to 2.79) | 0.64 |
| No. of COVID-19 symptoms | | | | |
| 0–5* | 52 | 1 | | |
| 6–9 | 78 | 1.32 | (0.57 to 3.05) | 0.52 |
| 10–23 | 102 | 1.29 | (0.55 to 3.01) | 0.56 |

* Baseline category

All tests but RVP showed age-related decline (p<0.05 for DMS, p<0.001 for OTS and SWM). OTS and RVP were associated with degree of education with better scores among those with the highest level of education (p<0.001), whereas DMS and SWM were not. No other variables significantly affected the cognitive outcomes in the four models.

The sensitivity analysis showed weighted percentages of subjects with a z-score <-1.5 for DMS of 13.5%, OTS 8.1%, RVP 3.4% and SWM 9.4%. The weighted logistic and multiple linear regression analyses showed similar patterns of coefficients as the primary analyses and did not alter the inferences or conclusions.

## Discussion

The major findings in this study were that cognitive function in our sample of COVID-19-infected, non-hospitalized patients 8–13 months after the acute phase was only marginally poorer than expected based on comparison with normative UK data, and the effect sizes were small. The proportion of respondents with z-scores lower than -1.5 was similarly small, though with larger effects in *post hoc* analyses of executive function among older respondents. Except for SWM, we found no association between COVID-19 symptom severity or CNS symptomatology (headache, confusion) with overall or dimension-specific cognitive impairment in multivariable regression analyses.

Most other studies with cognitive testing after acute COVID-19 have had rather small samples, often of selected hospitalized or referred patients in the early recovery phase [26–29]. The follow-up time after COVID-19 in the present study is longer than in previous studies and at a point in time when most patients would be expected to have recovered.

The prevalence of impairments in this study at about 11 months are in line with the findings in 379 outpatients a mean of 7.6 months after diagnosis [12]. As in the present study, that study defined cognitive impairments as z-score <1.5 SD below norms, and reported that

**Table 5. Determinants of cognitive test scores.** Unstandardized beta coefficients with 95% confidence intervals and p-values, multiple linear regression analysis with bootstrapping.

| | Delayed matching to sample (DMS-PCAD, higher score is better) | | One touch stockings of Cambridge (OTS-PSFC, higher score is better) | | Rapid visual information processing (RVP_A', higher score is better) | | Spatial working memory (SW-MBE, higher score is worse) | |
|---|---|---|---|---|---|---|---|---|
| | Coef. | 95%CI | Coef. | 95%CI | Coef. | 95%CI | Coef. | 95%CI |
| Age, per decade | -0.15* | [-0.26,-0.03] | -0.59*** | [-0.91,-0.27] | 0 | [-0.01,0.00] | 2.18*** | [1.20,3.15] |
| Sex | | | | | | | | |
| Female | 0 | | 0 | | 0 | | 0 | |
| Male | 0.06 | [-0.29,0.42] | 0.63 | [-0.11,1.38] | 0.01 | [-0.01,0.02] | -0.15 | [-2.80,2.50] |
| Education | | | | | | | | |
| Primary school (<10 years) | 0 | | 0 | | 0 | | 0 | |
| Secondary school (10–12 years) | 0.43 | [-0.10,0.96] | 1.19 | [-0.66,3.04] | 0 | [-0.02,0.03] | -1.01 | [-6.27,4.26] |
| University, <4 years | 0.24 | [-0.36,0.85] | 2.13* | [0.20,4.07] | 0.03 | [-0.00,0.05] | -3.85 | [-9.30,1.60] |
| University, ≥4 years | 0.19 | [-0.39,0.77] | 2.50** | [0.63,4.37] | 0.04** | [0.01,0.07] | -1.88 | [-7.16,3.39] |
| Born in Norway | | | | | | | | |
| No | 0 | | 0 | | 0 | | 0 | |
| Yes | -0.16 | [-0.71,0.40] | 0.73 | [-0.46,1.92] | 0.01 | [-0.01,0.03] | -1.7 | [-5.09,1.68] |
| Comorbidities | | | | | | | | |
| 0 | 0 | | 0 | | 0 | | 0 | |
| 1 | 0.02 | [-0.35,0.39] | 0.32 | [-0.55,1.20] | 0.01 | [-0.01,0.02] | -2.29 | [-5.15,0.57] |
| 2 | -0.1 | [-0.55,0.35] | -0.86 | [-2.24,0.52] | -0.01 | [-0.04,0.01] | 1.71 | [-2.46,5.88] |
| ≥3 | -0.34 | [-0.88,0.21] | -0.8 | [-2.49,0.90] | 0 | [-0.03,0.03] | -1.17 | [-5.94,3.60] |
| Headache during acute COVID-19 | | | | | | | | |
| No | 0 | | 0 | | 0 | | 0 | |
| Yes | -0.01 | [-0.38,0.36] | 0.52 | [-0.44,1.47] | 0 | [-0.02,0.01] | -1.2 | [-4.42,2.03] |
| Confusion during acute COVID-19 | | | | | | | | |
| No | 0 | | 0 | | 0 | | 0 | |
| Yes | -0.05 | [-0.54,0.45] | -0.91 | [-2.24,0.41] | 0.01 | [-0.01,0.03] | -0.73 | [-4.29,2.83] |
| Seizures during acute COVID-19 | | | | | | | | |
| No | 0 | | 0 | | 0 | | 0 | |
| Yes | 0 | [-1.00,1.00] | 0.28 | [-1.10,1.65] | -0.02 | [-0.04,0.01] | -1.97 | [-6.74,2.80] |
| Smell loss during acute COVID-19 | | | | | | | | |
| No | 0 | | 0 | | 0 | | 0 | |
| Yes | 0.13 | [-0.23,0.49] | 0.13 | [-0.74,1.01] | -0.01 | [-0.02,0.01] | 0.65 | [-2.26,3.56] |
| No. of symptoms during acute COVID-19 (0–23 scale) | | | | | | | | |
| 0–5 | 0 | | 0 | | 0 | | 0 | |
| 6–9 | -0.14 | [-0.61,0.34] | -0.44 | [-1.50,0.62] | -0.02 | [-0.03,0.00] | 4.26* | [0.65,7.86] |
| 10–23 | -0.19 | [-0.69,0.30] | -0.5 | [-1.76,0.76] | -0.02 | [-0.04,0.00] | 3.38 | [-0.40,7.16] |
| EQ-5D Anxiety/Depression after 3 months | | | | | | | | |
| 1–2 (None/slight) | 0 | | 0 | | 0 | | 0 | |
| 3–4 (Moderate/severe) | -0.28 | [-0.75,0.18] | -0.76 | [-2.34,0.81] | 0 | [-0.02,0.02] | 3.34 | [-0.16,6.83] |
| N | 222 | | 223 | | 216 | | 223 | |
| Adj. R-squared | 0 | | 0.19 | | 0.14 | | 0.12 | |

* p<0.05

** p<0.01

*** p<0.001

5–15% of outpatients, and 13–39% of hospitalized patients had cognitive impairments on a battery of tests [12]. The highest prevalence of impairments were in the cognitive domains of memory encoding and processing speed [12]. Our findings also support the results from a web-based study, where non-hospitalized COVID-19 cases had 0.04–0.13 SD lower test scores than those that believed they had not been ill [11]. The majority of patients had cognitive testing 1–5 months after illness onset.

Following COVID-19, persistent cognitive symptoms were reported in 14 non-hospitalized young/middle-aged patients 48–142 days after symptom onset [27], in contrast with our findings from cognitive testing.

Recent studies of survivors 2–4 months after hospitalization for COVID-19 reported a high prevalence of moderate/severe neurocognitive impairment, as assessed by telephone [8] or a brief performance-best test battery [29], and telephone-administered cognitive tests were abnormal in 15–17% about 3 months after COVID-19 in a mixed population of hospitalized and referred patients [9]. Another study of hospitalized patients reported frequent cognitive abnormalities about 5 months after hospital discharge, in particular increased fatigability, and deficits of concentration and memory [30]. The latter finding corresponds with our finding of reduced function on the spatial working memory test (SWM), although the difference from the norm population was small in the present study.

Patients seeking neurological counseling for neurocognitive symptoms about 6 months after COVID-19 had only minor impairments at single-patient level [31]. Similarly, cognitive deficits were reported from selected patients following COVID-19 with initial neurologic symptoms lasting over 6 weeks [28], in 100 patients in a population with 70% females, and 85% reporting fatigue.

A longitudinal study with available cognitive scores 1–7 years before COVID-19 reported cognitive decline in middle-aged and old adults with mild symptomatic COVID-19 [32]. In contrast, another study did not show a difference in global cognition scores at 4 months between COVID-19-affected and non-affected health care workers [10].

Our findings complement these more short-term observational studies and suggest that most patients infected by COVID-19 but not requiring hospital admission, show little cognitive impairments after 8–13 months.

In another study, headache, oxygen therapy, anosmia and dysgeusia were associated with cognitive impairment following hospitalization for COVID-19 [25], though the sample was small and the analysis limited to univariate analysis. This contrasts with the finding in the present, larger study of no associations between headache or other initial COVID-19 symptoms and cognition, in a non-hospitalized population with less severe COVID-19, longer follow-up before cognitive assessments, and multivariable analysis.

We used tests from the CANTAB battery because we expected that shorter cognitive tests, such as the Mini Mental State Exam or MoCa would be less sensitive to capture potential, possibly dimension-specific deficits, and that traditional neuropsychological assessment would require more resources than were available. We cannot exclude the possibility that traditional neuropsychological tests may have been more sensitive to COVID-19-induced changes, as the CANTAB only shows modest associations with traditional neuropsychological test measures [33].

The main weaknesses of the study are that we had limited information about pre-infection cognitive symptoms or functioning, and the lack of a feasible comparison group. Most of the patients, however, had been working full time before COVID-19, and the prevalence of comorbidities was low. In addition, the sample size may seem small. However, in terms of the total number of COVID-19-verified non-hospitalized patients available in these districts in Norway at the time, and in comparison with the size of other similar studies, we think the sample size

is sufficient to give important and reasonably generalizable results. We naturally also cannot exclude a role of other, as yet, unidentified confounders.

From available CANTAB tests, we chose those available in Norwegian, as well as with population norms. The chosen battery is very similar to that recently recommended for studies following COVID-19 [19]. Choice of comparison group in cognitive testing remains a challenge for interpretation of findings following COVID-19. Very few studies would have pre-COVID-19 cognitive tests for the same individuals, and in case, this would probably be because there was a special indication. Other options would be to have a non-COVID-19 group from the general population, or possibly hospitalized for another reason for comparison. The choice of control group could in any case be criticized, and there may be proponents of different choices. We think we used the best option that was available to us. As no local normative data is available, we used UK norms provided by the vendor [22]. The norms were derived from web-based cognitive assessment from a UK population ≥18 years of age, with no previous significant head injury (resulting in loss of consciousness), no mental health condition that is uncontrolled (by medication or intervention) and which has a significant impact on daily life, and no previous diagnosis of mild cognitive impairment or dementia [22]. We suggest this to be a reasonable comparator when addressing a pandemic that affects the general population. The CANTAB system has been used in numerous studies in many countries [33].

We think the pattern of better cognitive scores with declining age and increasing length of education in the regression analysis on raw scores support the validity of the tests as do perhaps the *post hoc* analyses suggesting a reduced executive function in the older portion of the sample.

This study was originally population-based and reasonably representative of the population of non-hospitalized patients with COVID-19. The respondents in the first round were older (mean age on 1 June 2020 49.5 (SD 15.3) vs. 43.9 (17.3) years for non-respondents (p<0.001) and comprised a larger proportion of women (256 (56%) vs. 219 (46%), p = 0.005) [34]. The response rates in seven aggregated geographical districts within the catchment areas of the hospitals ranged from 26% to 65%, with lowest rates in three districts of Oslo that have a large proportion of immigrants in the population [15]. This may have introduced bias in the sample.

Furthermore, during the clinic visit in the present study we did not conduct cognitive tests in those with dyspnea only and under-sampled those without dyspnea or fatigue for the cognitive testing, which possibly could have contributed to lower overall cognitive scores. Respondents to CANTAB had more baseline symptoms and lower quality of life, but did not differ from non-respondents in other characteristics. This may have caused a selection bias and may limit generalization. However, we do not know to what extent this may influence associations between cognitive test scores and the chosen determinants. The sensitivity analyses, taking this into consideration, did not materially influence the results and the inferences.

The lack of major late cognitive reduction in our sample of non-hospitalized, but PCR-verified COVID-19 patients seems to advocate against SARS-Cov-2 viral infection of the CNS itself as being causative in cognitive changes after the acute disease phase.

If cognitive sequelae were associated with hypoxia concomitant with cerebrovascular disease or due to critical illness, this would be expected to be more severe in hospital-admitted or ICU-treated cases that were not included in the present study. Most serious indirect causes of damage would probably have led to hospital admission or be partly a result of intensive hospital treatment.

In conclusion, this study of non-hospitalized patients 8–13 months after COVID-19 found only slightly lower cognitive scores in this population compared to available population norms. This is reassuring and suggests that most non-hospitalized patients have little cognitive sequela following the infection, despite a multitude of symptoms in the acute phase.

## Supporting information

**S1 Table. Non-response analysis among those that responded to the first survey (n = 458).** Comparison between those completing CANTAB assessment during the follow-up visit (n = 233) and non-participants/not responding to CANTAB (n = 225).
(PDF)

**S2 Table. Cognitive test scores for subjects ≥60 years and comparison with norm population.**
(PDF)

## Author Contributions

**Conceptualization:** Knut Stavem, Gunnar Einvik, Waleed Ghanima, Erik Hessen, Christofer Lundqvist.

**Data curation:** Knut Stavem, Gunnar Einvik, Birgitte Tholin.

**Formal analysis:** Knut Stavem, Christofer Lundqvist.

**Investigation:** Knut Stavem, Gunnar Einvik, Waleed Ghanima, Erik Hessen.

**Methodology:** Knut Stavem, Gunnar Einvik, Birgitte Tholin, Waleed Ghanima, Erik Hessen, Christofer Lundqvist.

**Project administration:** Knut Stavem, Gunnar Einvik, Birgitte Tholin, Waleed Ghanima.

**Resources:** Waleed Ghanima.

**Supervision:** Gunnar Einvik, Birgitte Tholin, Waleed Ghanima, Erik Hessen.

**Validation:** Knut Stavem, Gunnar Einvik, Erik Hessen, Christofer Lundqvist.

**Visualization:** Knut Stavem.

**Writing – original draft:** Knut Stavem, Christofer Lundqvist.

**Writing – review & editing:** Knut Stavem, Gunnar Einvik, Birgitte Tholin, Waleed Ghanima, Erik Hessen, Christofer Lundqvist.

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
