## [Decision Letter · Decision Letter 0]

30 May 2022

PONE-D-22-12572Cognitive function in non-hospitalized patients 8–13 months after acute COVID-19 infection: a cohort studyPLOS ONE

Dear Dr. Stavem,

Thank you for submitting your manuscript to PLOS ONE. After careful consideration, we feel that it has merit but does not fully meet PLOS ONE’s publication criteria as it currently stands. Therefore, we invite you to submit a revised version of the manuscript that addresses the points raised during the review process.

We look forward to receiving your revised manuscript.

Kind regards,

Tai-Heng Chen, M.D.

Academic Editor

PLOS ONE

a) Did participants provide their written or verbal informed consent to participate in this study?

Reviewers' comments:

Reviewer's Responses to Questions

**Comments to the Author**

1. Is the manuscript technically sound, and do the data support the conclusions?

Reviewer #1: Yes

Reviewer #2: Yes

2. Has the statistical analysis been performed appropriately and rigorously? 

Reviewer #1: Yes

Reviewer #2: Yes

3. Have the authors made all data underlying the findings in their manuscript fully available?

Reviewer #1: No

Reviewer #2: Yes

4. Is the manuscript presented in an intelligible fashion and written in standard English?

Reviewer #1: Yes

Reviewer #2: Yes

5. Review Comments to the Author

Reviewer #1: The authors investigated the cognitive function for non-hospitalized patients with PCR-confirmed COVID-19 patients. They longitudinally followed them.

At median 11 (range 8–13) months after PCR positivity, cognitive scores for short term memory, visuospatial processing, learning and attention were lower than norms (p≤0.001).

At median 11 months after out-of-hospital SARS-Cov-2 infection, minor cognitive impairment was seen with little association between COVID-19 symptom severity and outcome.

I will raise some concerns before publication.

Major

#1

Why did not the authors perform cognitive testing for dyspnea group? The authors hypothesized that the cognitive sequelae were associated with hypoxia. Then, dyspnea group should be conducted cognitive testing.

Could you elaborate on “Because of risk of attrition”?

#2

As the authors pointed out, the repone rate was low. Selection bias should be considered.

Are there any differences between response and non-response to questionnaire of survey 1?

If any, discuss it. And are there any differences between response and non-response to CANTAB of survey 2? If any, discuss it.

Minor

#3

In Result, Sample

“191 of these patients refused further follow-up” should be “19 of these patients refused further follow-up”.

#4

Please mention the COVID-19 infectious situation, such as local infections prevalence and mutation of prevalent SARS-CoV-2.

Over all, the paragraph division is bad and it is difficult to read.

Reviewer #2: Overall statement

This study aims to evaluate the effect of Covid-19 infection on cognitive function in non-hospitalized patients. The researchers found that late after a Covid-19 infection, non-hospitalized patients showed only slightly lower cognitive test scores. The research question is clearly outlined. The authors used a detailed cognitive test battery. The results seem to be valid and reliable. In my opinion the sample size is limited for a population based study.

Strengths and impact

This is a longitudinal study that includes evaluations with a good quality cognitive test battery (CANTAB). This battery evaluates many cognitive domains, especially executive functions. The title of the study is informative. The study aim is interesting since they measured the cognitive effects after a Covid-19 infection in a non-hospitalized group with low comorbidity scores. The references at the end of the text are up to date. The current knowledge about the topic are given in detail. Statistical methodology seems to be nice (using Z scores). The manuscript is generally well written (especially discussion).

Minor weaknesses

1. Even if they designed a longitudinal study, the reserchers very found mild differences in cognitive scores (%4-14). Since this is a cognitive study. The results must be discussed further in relation to participant age, educational status and comorbid conditions. It may be possible to find significant cognitive differences in an elderly population with a larger sample size.

2. The introduction is well written. The author wrote that they aimed to determine the prevalence of cognitive deficits 8-13 months after Covid-19 infection. The author must explain the importance/meaning of this time interval (8-13 months) since they wrote it in the study aim and the title.

3. Did any of the patients have CT documented pulmonary Covid disease? The Figure legends are missing.

4. The researcher divided the patients with a Covid-19 history into 3 groups according to their symptoms (fatigue, dyspnea & fatigue, no dyspnea or fatigue). I think that this classification makes the study a little bit confusing. If I were to do a patient classification for non-hospitalized Covid-19 patients I would divide them into 2 groups such as; asymptomatic/mild symptomatic and symptomatic (including dyspnea and apparent fever). Maybe if they made 2 patient groups with a different methodology, they could have included dyspnea only patients and the remaining asymptomatic patients to increase the sample size.

5. Table 1. is informative. Table 3. is easy to understand and definitive. The regression analysis did not reveal any significance except patient age (Table 5.). The author may want to make a subgroup analysis in older participants to see which cognitive domains were effected after an acute Covid-19 infection.

6. In the results/sample section the number of patients participated in CANTAB was given as 233. In Table 2. the number seems to be 234?

7. Even if the author made adjustments for possible confounders, there may be other confounders not taken into account. There is no control group as the author mentioned. Therefore, it would be better if the author writes some more possible limitations.

Major weaknesses

1. In my opinion the sample size is limited for a population based study.

6. PLOS authors have the option to publish the peer review history of their article (what does this mean?). If published, this will include your full peer review and any attached files.

Reviewer #1: No

Reviewer #2: No

---

## [Decision Letter · Decision Letter 1]

12 Jul 2022

PONE-D-22-12572R1Cognitive function in non-hospitalized patients 8–13 months after acute COVID-19 infection: a cohort studyPLOS ONE

Dear Dr. Stavem,

Thank you for submitting your manuscript to PLOS ONE. After careful consideration, we feel that it has merit but does not fully meet PLOS ONE’s publication criteria as it currently stands. Therefore, we invite you to submit a revised version of the manuscript that addresses the points raised during the review process.

We look forward to receiving your revised manuscript.

Kind regards,

Tai-Heng Chen, M.D.

Academic Editor

PLOS ONE

Journal Requirements:

Reviewers' comments:

Reviewer's Responses to Questions

**Comments to the Author**

1. If the authors have adequately addressed your comments raised in a previous round of review and you feel that this manuscript is now acceptable for publication, you may indicate that here to bypass the “Comments to the Author” section, enter your conflict of interest statement in the “Confidential to Editor” section, and submit your "Accept" recommendation.

Reviewer #1: All comments have been addressed

Reviewer #2: All comments have been addressed

2. Is the manuscript technically sound, and do the data support the conclusions?

Reviewer #1: Yes

Reviewer #2: Yes

3. Has the statistical analysis been performed appropriately and rigorously? 

Reviewer #1: Yes

Reviewer #2: N/A

4. Have the authors made all data underlying the findings in their manuscript fully available?

Reviewer #1: No

Reviewer #2: Yes

5. Is the manuscript presented in an intelligible fashion and written in standard English?

Reviewer #1: Yes

Reviewer #2: Yes

6. Review Comments to the Author

Reviewer #1: The manuscripts were well revised.

All my concerne were satisfied.

I raise one addition comments.

Minor

The title should be revised,

Cognitive function in non-hospitalized patients 8–13 months after acute COVID-19

infection: a cohort study

===>

Cognitive function in non-hospitalized patients 8–13 months after acute COVID-19

infection: a cohort study in Norway.

Reviewer #2: This is the revised manuscript of a study that aimed to evaluate the effect of Covid-19 infection on cognitive function in non-hospitalized patients. In my opinion the revisions are OK. Author responses are satisfactory.

7. PLOS authors have the option to publish the peer review history of their article (what does this mean?). If published, this will include your full peer review and any attached files.

Reviewer #1: No

Reviewer #2: No

---

## [Decision Letter · Decision Letter 2]

8 Aug 2022

Cognitive function in non-hospitalized patients 8–13 months after acute COVID-19 infection: a cohort study in Norway

PONE-D-22-12572R2

Dear Dr. Stavem,

We’re pleased to inform you that your manuscript has been judged scientifically suitable for publication and will be formally accepted for publication once it meets all outstanding technical requirements.

Kind regards,

Tai-Heng Chen, M.D.

Academic Editor

PLOS ONE

Reviewers' comments:

Reviewer's Responses to Questions

**Comments to the Author**

1. If the authors have adequately addressed your comments raised in a previous round of review and you feel that this manuscript is now acceptable for publication, you may indicate that here to bypass the “Comments to the Author” section, enter your conflict of interest statement in the “Confidential to Editor” section, and submit your "Accept" recommendation.

Reviewer #1: All comments have been addressed

2. Is the manuscript technically sound, and do the data support the conclusions?

Reviewer #1: Yes

3. Has the statistical analysis been performed appropriately and rigorously? 

Reviewer #1: Yes

4. Have the authors made all data underlying the findings in their manuscript fully available?

Reviewer #1: Yes

5. Is the manuscript presented in an intelligible fashion and written in standard English?

Reviewer #1: Yes

6. Review Comments to the Author

Reviewer #1: The title has been changed as a comment.

All my concerns were satisfied.

7. PLOS authors have the option to publish the peer review history of their article (what does this mean?). If published, this will include your full peer review and any attached files.

Reviewer #1: No

---

## [Editor Report · Acceptance letter]

11 Aug 2022

PONE-D-22-12572R2 

Cognitive function in non-hospitalized patients 8–13 months after acute COVID-19 infection: a cohort study in Norway 

Dear Dr. Stavem:

I'm pleased to inform you that your manuscript has been deemed suitable for publication in PLOS ONE. Congratulations! Your manuscript is now with our production department. 

Kind regards, 

on behalf of

Dr. Tai-Heng Chen 

Academic Editor

PLOS ONE